# RAD51C-XRCC3 structure and cancer patient mutations define DNA replication roles

**Michael A. Longo[1,6], Sunetra Roy[2,6], Yue Chen[2], Karl-Heinz Tomaszowski[2], Andrew S. Arvai[3], Jordan T. Pepper[4], Rebecca A. Boisvert[2], Selvi Kunnimalaiyaan[2], Caezanne Keshvani[2], David Schild[5], Albino Bacolla[1], Gareth J. Williams[4] ✉, John A. Tainer ◉[1,2] ✉ & Katharina Schlacher ◉[2] ✉**

*RAD51C* is an enigmatic predisposition gene for breast, ovarian, and prostate cancer. Currently, missing structural and related functional understanding limits patient mutation interpretation to homology-directed repair (HDR) function analysis. Here we report the RAD51C-XRCC3 (CX3) X-ray co-crystal structure with bound ATP analog and define separable RAD51C replication stability roles informed by its three-dimensional structure, assembly, and unappreciated polymerization motif. Mapping of cancer patient mutations as a functional guide confirms ATP-binding matching RAD51 recombinase, yet highlights distinct CX3 interfaces. Analyses of CRISPR/Cas9-edited human cells with *RAD51C* mutations combined with single-molecule, single-cell and biophysics measurements uncover discrete CX3 regions for DNA replication fork protection, restart and reversal, accomplished by separable functions in DNA binding and implied 5' RAD51 filament capping. Collective findings establish CX3 as a cancer-relevant replication stress response complex, show how HDR-proficient variants could contribute to tumor development, and identify regions to aid functional testing and classification of cancer mutations.

Sequence analysis of 1100 families with breast and ovarian cancer history, but without *BRCA1/2* mutations, uncovered *RAD51C* as a cancer predisposition gene[1]. Since then, mutations in *RAD51C* have been identified in many cancers, particularly those of endocrine organ origins including prostate[2], with variants of unknown significance (VUS) mapping throughout the protein sequence[3]. Furthermore, *RAD51C* is a Fanconi Anemia suppressor gene, a rare recessive genetic disorder resulting in hematological and developmental defects and cancer, with the same *RAD51C* mutation found in both Fanconi Anemia and breast cancer patients[1,2,4,5]. RAD51C is a central member of the RAD51 paralog family of proteins that share sequence homology to RAD51, the recombinase during homology-directed DNA double-strand break repair (HDR), and with incompletely understood functions in this

pathway downstream of the classical cancer susceptibility genes *BRCA1/2*[6]. RAD51C also is a potential biomarker for BRCA-related targeted therapies such as PARP inhibition[7]. Despite its importance in cancer development and therapy response, RAD51C functional studies have proven difficult due to its low cellular abundance, protein instability, embryonic lethality when deleted in mice, and essentiality in primary and most cancer cells[3,8,9]. Inferring from protein sequence, RAD51C contains Walker A/B ATP binding motifs, as well as a nuclear localization sequence at the C-terminal end (Fig. 1a).

ATP binding is essential for protein interaction and complex formation with other RAD51 paralogs[6,10,11]. RAD51C binds XRCC3 (CX3) to form a heterodimer, and separately with RAD51B to combine with the RAD51D-XRCC2 dimer into a hetero-tetramer (BCDX2)[10,11], whereby all

[1]Department of Molecular & Cellular Oncology, UT MD Anderson Cancer Center, Houston, TX, USA. [2]Department of Cancer Biology, UT MD Anderson Cancer Center, Houston, TX, USA. [3]The Department of Integrative Structural & Computational Biology, The Scripps Research Institute, La Jolla, CA, USA. [4]Department of Biochemistry and Molecular Biology, Robson DNA Science Centre, Charbonneau Cancer Institute, Cumming School of Medicine, University of Calgary, Calgary, AB, Canada. [5]Life Sciences Division, Lawrence Berkeley National Laboratory, Berkeley, CA 94720, USA. [6]These authors contributed equally: Michael Longo, Sunetra Roy. ✉e-mail: gareth.williams2@ucalgary.ca; jtainer@mdanderson.org; kschlacher@mdanderson.org

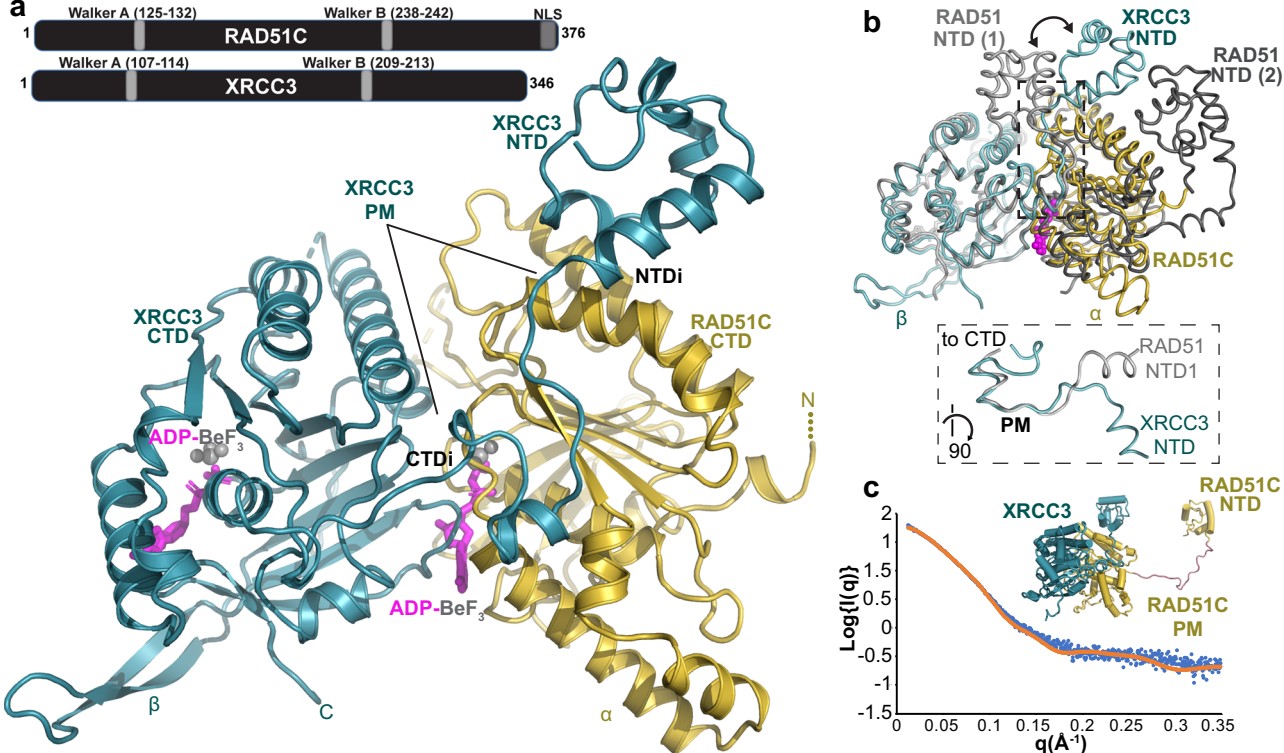

**Fig. 1 | RAD51C-XRCC3 X-ray crystal structure and X-ray scattering define conserved fold and assembly. a** Overview of the apCX3 heterodimer structure with important subdomains, regions, unique features for RAD51C (gold) and XRCC3 (teal), and bound ATP-mimic (magenta) highlighted. top, sketch of RAD51C and XRCC3 sequence with known motifs. **b** Structural superposition of the apCX3 core dimer with ATP-bound human RAD51 dimer (PDB 7EJC) to highlight the unique positioning of the XRCC3 NTD (teal) subdomain in the CX3 structure (see arrows). Dashed box, zoomed view of boxed area, rotated 90°, comparing the polymerization motif (PM) of RAD51 and XRCC3. **c** Comparison of a structural model of full-length hCX3 based on our apCX3 core and apRAD51C NTD structures to experimental SAXS data collected on hCX3.

paralogs except RAD51B are considered essential for cellular survival[3]. RAD51 paralogs have implied roles in RAD51 loading and filament stabilization functions that resemble those of BRCA2, so initial functional studies focused on homology-directed DNA double-strand break repair (HDR) functions, where BCDX2 has early and CX3 late functions in RAD51 filament assembly[6]. Also, putative Holliday junction resolvase activity within the N-terminal region of RAD51C was proposed[12], albeit other recombination intermediate resolvases have been identified since. So RAD51 paralog-interaction and HDR proficiency is currently used for functional testing of patient variants, which are criteria considered in variant pathogenicity predictions. Yet, recently RAD51C replication functions distinct from HDR functions were described, including replication fork protection, fork restart and fork reversal[13–16]. Fork protection requiring RAD51 filament stabilization[17] is moreover consistent with reported RAD51 filament remodeling and stabilizing properties of nematode paralogs[18].

Here we describe X-ray crystal and in solution X-ray scattering structures of RAD51C and its RAD51C-XRCC3 complex that reveal structurally distinct interfaces between these paralogs compared to RAD51 and RecA recombinases. Informed by recurrent cancer mutations mapped upon the atomic structure combined with single-molecule and single-cell measurements, the results uncover regions of distinct replication stress roles that together provide foundational biological knowledge and a framework for targeted cancer testing mutations in patients.

## Results

### Co-crystal structure of RAD51C-XRCC3 (CX3)

Purified human CX3 (hCX3) typically yields aggregated and inactive protein complexes[19]. We therefore sequenced and identified

conserved *RAD51C* (GenBank: OQ586109) and *XRCC3* (GenBank: OQ586109) genes in *Alvinella pompejana* (ap), an extreme metazoan with thermostable proteins[20], which share high amino acid similarity and identity to human RAD51C (80% similarity, 56% identical) and XRCC3 (74% similarity, 43% identical) (Supplementary Fig. 1). We designed several apRAD51C, apCX3 and hCX3 constructs (Supplementary Fig. 2a). Limited proteolysis analysis of hCX3 suggested that a small RAD51C N-terminal domain (NTD) is flexibly attached to the CX3 complex (Supplementary Fig. 2b). Consistently, crystals of apCX3 with both full-length proteins diffracted to a low resolution and determined preliminary structures lacked electron density for the RAD51C NTD. Using purified apCX3 core complex without the resolution-limiting RAD51C NTD, we solved the 2.6 Angstrom (Å) X-ray crystal structures of CX3 complexed with ADP-BeF₃, a non-hydrolysable ATP mimic, bound in the active sites of both RAD51C and XRCC3 (Fig. 1a, Supplementary Figs. 2c, d and 3, and Supplementary Table 1). In addition, we solved high-resolution crystal structures of apRAD51C NTD to 1.6 Å and nucleotide-bound apRAD51C CTD to 2.3 Å (Supplementary Fig. 2e, f and Supplementary Table 1). Superposition of the apRAD51C CTD with the apCX3 complex show minimal structural changes, so our analysis here focused on the biologically relevant CX3 complex. Importantly, apCX3 crystallized with three independent dimers (totaling 6 chains and over 1,800 residues) in the asymmetric unit and conserved superpositions. We therefore were able to fit and refine three independent structures of the CX3 heterodimer into the electron density maps: this provides high confidence in the defined interfaces and conformations.

The overall apCX3 structure shows extended interfaces between RAD51C and XRCC3 with extensive contacts between the RAD51C CTD and the XRCC3 NTD, CTD, and a distinct linker polymerization motif

that connects the subdomains (Fig. 1a and Supplementary Fig. 2g). In all three heterodimers, the structures reveal a conserved CX3 ATPase dimer interface mediated by the bound ATP transition state analog. ADP·BeF₃ is buried deep within the complex sandwiched between CX3 subunits, with the RAD51C Walker motifs coordinating the BeF₃ phosphate mimic. This nucleotide binding matches that seen in crystal structures of inactive RecA and RAD51[21]. Likewise, two lysine side chains charge-stabilize the terminal BeF₃ group such that ATP hydrolysis and release of inorganic phosphate could in principle destabilize the active CX3 interface by introducing a net charge in a buried environment, jeopardizing complex stability. Notably, during optimization of protein purification we found the addition of ATP, Mg²⁺ and vanadate (a phosphate mimic that traps a nucleotide bound state of the complex) prevents aggregation of hCX3 (Supplementary Fig. 2h, i).

Yet, apCX3 strikingly differs from published structures of RecA and RAD51 family members with the unique positioning of the XRCC3 NTD, which is rotated ~90 degrees compared to RAD51-RAD51 interactions. This repositioning of the NTD results from a unique XRCC3 linker protein-polymerization motif, with residues at the N-terminal region of this motif driving the NTD reorientation compared to RAD51 (Fig. 1b). However, this XRCC3 motif also shares features of both the RecA and RAD51 polymerization motifs at the C-terminal interaction region, albeit with a six amino acid insertion (Supplementary Fig. 2e). The reorientation of the XRCC3 NTD creates a structure in which XRCC3 resembles a C-clamp, with the two sides of the clamp formed by the XRCC3 CTD-RAD51C CTD (CTDI) interface and the unique XRCC3 NTD-RAD51C CTD interface on another surface of RAD51C (NTDI). This interaction modality, which is enabled by the unique XRCC3 linker polymerization motif, increases CX3 interface area compared to RAD51 dimers. Added distinct elements of an extended α-hairpin insertion in RAD51C (Fig. 1a, b hRAD51C residues 179–199) and a protruding β-hairpin insertion in XRCC3 (apXRCC3 residues 292–310) that in computational models of human XRCC3 structure is predicted as a short helix (residues 300–307), suggest potential additional interaction interfaces.

To ensure that the apCX3 structure is generally representative of the evolutionarily conserved CX3, we performed small-angle X-ray scattering (SAXS) of near full-length hCX3 (RAD51C residues 10–367, full-length XRCC3), which informs on the structure in solution including flexible conformations (Fig. 1c and Supplementary Fig. 2a, Supplementary Table 2). In analysis with BILBOMD[22], a structural model of full-length hCX3 based on our apCX3 and apRAD51 NTD crystal structures matches hCX3 in solution with excellent fit (Chi² = 1.32). This data is consistent with the apCX3 core complex adopting the same structure as the human complex in solution with the RAD51C NTD flexibly attached, in concordance with our limited proteolysis results (Fig. 1c and Supplementary Fig 2b). Thus, the X-ray crystal structure, sequence conservation, and X-ray scattering measurements establish apCX3 as a reliable surrogate for human CX3 structure and assembly.

The combined crystallographic structures plus SAXS data on CX3 in solution furthermore show both conserved and unique features compared to RAD51. Notably, based on sequence conservation and shown by our structures, the NTD is generally expected to be similar between RAD51 and the paralogs. Yet, the unanticipated positioning of XRCC3 NTD by the unique linker polymerization motif forms a unique interface with RAD51C, and these differences are expected to be responsible for distinct CX3 biological activities compared to RAD51.

## Cancer patient mutations at interaction regions

*RAD51C* cancer mutations occur throughout the protein sequence, challenging prediction of functional impacts and implied pathogenicity. *RAD51C* was identified as an inherited cancer susceptibility gene by sequencing of families with history of breast and ovarian cancer that revealed 11 variants[1]. In addition, a RAD51C variant R258H was

identified in a Fanconi Anemia patient[4], and is also found in cancer patients[2]. In general, variants originally identified in breast/ovarian cancer patients were later also found in Fanconi Anemia patients, and vice versa. This observation suggests that the same mutational defects can give rise to these related but distinct diseases due to additional factors[2,5]. Functional analysis for HDR of the originally identified cancer and Fanconi Anemia variants revealed three defective variants, which served as confirmation for classifying them as pathogenic (G125V, L138F, and R258H)[1,4]. The remaining variants[1] were classified likely benign at least in part driven by the apparent absence in functional defects as measured by HDR proficiency. Nevertheless, these variants were recurring in familial and sporadic breast, ovarian and diverse cancer studies[23–30], which resulted in conflicting interpretation with a remaining degree of uncertainty. We reasoned that these variants may reveal functional importance irrespective of HDR proficiency motivating our efforts to define their potential structural and functional roles.

The CX3 structure reveals two distinct areas in three dimensions to which the mutations map (Fig. 2a). R258H, which is pathogenic[4], and uncertain significance variant G264S[1], map to a RAD51C α-helix central to the XRCC3 NTDI (Fig. 2b). The CX3 structure shows that G264 is within 4.5 Å of the XRCC3 NTD, with combined electrostatic and hydrophobic interactions mediating this unique interface. R258, neighboring the XRCC3 NTD, maps to the end of the α-helix that also contains G264, suggesting likely functionally related impact of the G264S and R258H mutations.

In contrast to the NTDI, pathogenic mutations G125V, L138F, and likely benign/uncertain significance variants A126T, D159N, V169A and T287A map within the core of the CX3 complex neighboring the ATPase active site and CTDI (Fig. 2c). Notably, the molecular environment of the CX3 CTDI extends beyond the ADP·BeF₃ binding site described above. Interestingly, all of the RAD51C pathogenic mutations and VUS's that map across the CTDI have structural connections to ATP binding or hydrolysis and/or XRCC3 CTD binding, despite not being directly part of the Walker A/B motif residues previously noted as important for CX3 interaction[11]. The RAD51C G125V and A126T neighbor both the ATP binding site and an XRCC3 interface. Similarly, D159N maps to a region adjacent to the Walker B ATPase motif and next to RAD51C residues that interact with XRCC3. L138F and V169A are in the RAD51C core on α-helices that make important connections to the nucleotide and the Walker B motif (L138F) or ATP and the XRCC3 CTD (V169A). In the protein sequence, T287A is far removed from Walker A/B motifs. Yet on the three-dimensional structure, it maps to a loop with several other RAD51C residues that interact with XRCC3 to anchor the CX3 CTDI.

Despite decades of effort by many groups, CX3 variants have proven technically difficult to rigorously study either biochemically (due to the formation of mixed aggregated and filament complexes) or in cells (due to confounding impacts of non-endogenous expression levels on any assessment of functional defects). With these points in mind and the patient variants defining CTDI and NTDI on the CX3 structures, we sought to obtain a better functional understanding of the two distinct interaction regions. Full-length *RAD51C* knock-out is lethal in normal and cancer cells with some exceptions[3,8,9], and functional RAD51C variant testing so far relied on knockdown of the protein or variant overexpression in cells that have overcome RAD51C essentiality, which may have secondary effects. We therefore used human isogenic, CRISPR-Cas9 edited HAP1 cells with endogenously expressed RAD51C A126T (affecting CTDI) or G264S (affecting NTDI) variants compared to wild-type RAD51C expressing cells (Supplementary Fig. 4a). We reasoned that endogenous expression provides exemplary tests for these two interface regions and may prove more sensitive and informative than data from multiple variants with varying degrees of overexpression. In contrast to *RAD51C* knock-out[9], HAP1 cells containing the patient variant mutations are viable.

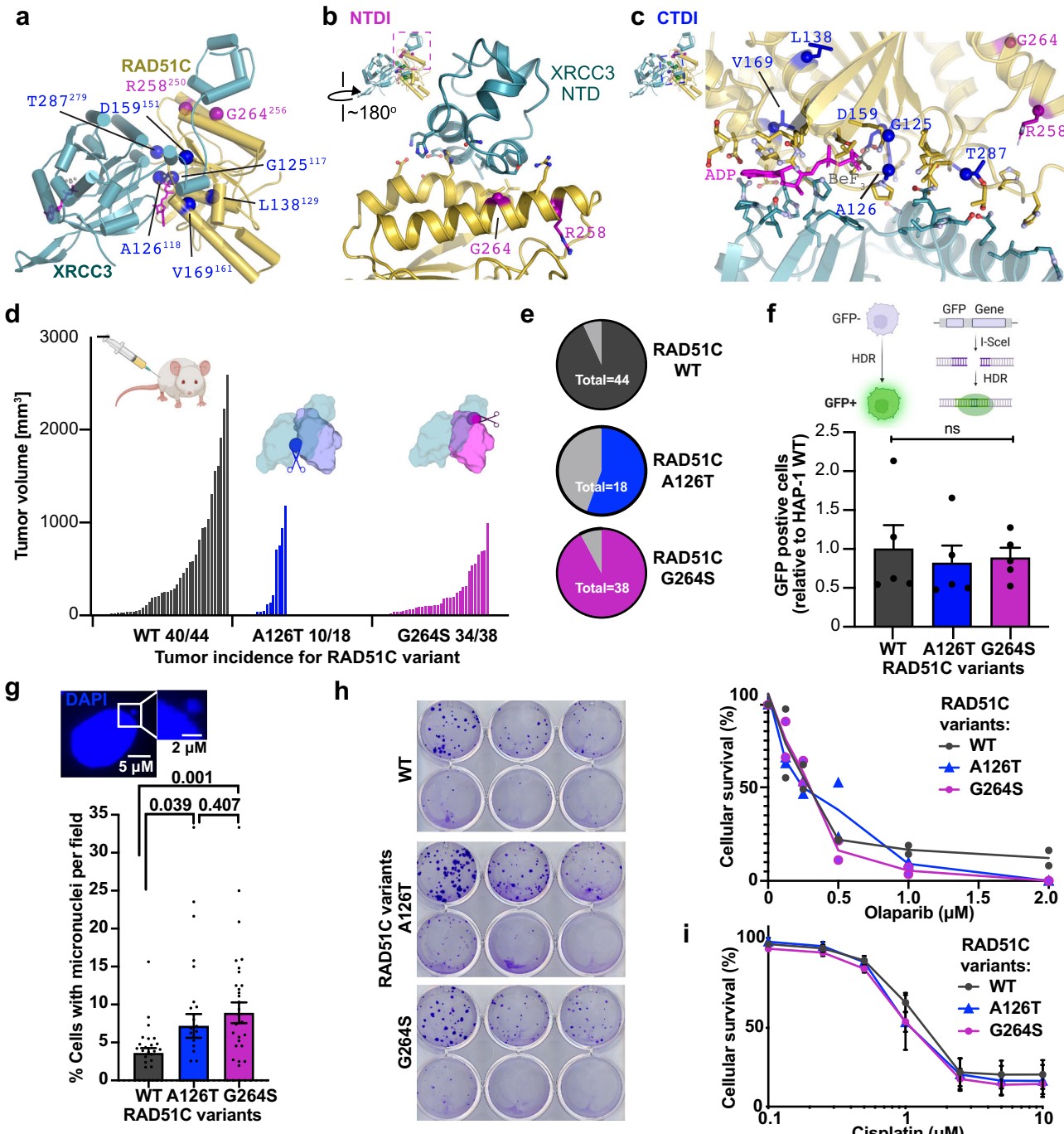

**Fig. 2 | RAD51C cancer mutations spotlight two RAD51C-XRCC3 interface regions. a** Mapping of recurrent RAD51C cancer mutations onto the apCX3 structure. Mutations clustered at the RAD51C-XRCC3 NTD interface with RAD51C (NTDI) are colored purple and mutations mapping around the CX3 ATP and CTD interface (CTDI) are colored blue. Residue numbering denotes human RAD51C with corresponding *A. pompejana* numbering in superscript. **b** Close up view to show residues within 4.5 Å of the RAD51C-XRCC3 NTDI. **c** Close up view of the extended CX3 CTDI. Residues within 4.5 Å of the interface are shown. **d** Tumor volume for xenografts of HAP1 cells expressing wild-type (WT) or RAD51C variants as indicated. Labeling on the x-axis denotes the number of tumors that grew compared to the number of independent injections. Schematics created with http://BioRender.com. **e** Pie chart representation of tumor take rate. **f** Dr-GFP HDR proficiency assays of HAP1 cells expressing wild-type (WT) or variant RAD51C as indicated. Top, schematic created with http://BioRender.com of Dr-GFP assay with I-Sce1 endonuclease induced double-strand break to measure homology directed repair (HDR) efficiency by the restoration of a functional green fluorescence protein (GFP). Data are presented as the mean +/− SEM. *P* values were calculated using an unpaired, two-sided Student T-test, *n* = 5 independent biological experiments. **g** Wild-type or RAD51C variant expressing HAP1 cells were assessed for presence of micronuclei expressed as percentage of cells with micronuclei per image field. Data are presented as the mean +/− SEM. *P* values were calculated using an unpaired, two-sided Student T-test, *n* = 5 independent biological experiments resulting in a total of n(WT) = 1351, n(A126T) = 1003, and n(G264S) = 1090. **h** Clonogenic survival assay with Olaparib. Left, images of plates exposed to varying concentrations of Olaparib increasing in from 0 at the top left to 2 μM at the bottom right. Right, Quantification whereby error bars represent the standard error of the mean, *n* = 2 independent biological experiments. **i** MTS survival assay with cisplatin. *N* = 5 independent biological experiments performed in triplicates, error bars represent the standard error of the mean.

We tested cellular proliferation of the cells by injecting RAD51C wild-type, A126T, or G264S HAP1 cells into the hind-legs of nude mice as a more sensitive and biologically relevant means of testing cell growth (Fig. 2d). While xenograft take rates are lower for cells with A126T compared to G264S mutations, both mutant cells showed significantly decreased tumor volumes compared to cells containing wild-type RAD51C (Fig. 2d, e). This result suggests a significant functional impact of the mutations on tumor proliferation, which is a common consequence also of HDR deficiency seen in xenografts with BRCA2 defective cells[31,32]. In addition, endogenous micronuclei signifying unresolved replication fork stress are significantly increased in cells with either RAD51C A126T or G264S compared to wild-type RAD51C (Fig. 2g)

In previous tests, overexpression of these variants in diverse RAD51C defective cells did not change or only mildly affect HDR proficiency[1,3,33]. Consistently, in our isogenic cell lines with endogenous expression of either G264S or A126T, HDR capacities of the cells are not significantly changed as measured by the DR-GFP assay. This assay measures the restoration of an unfunctional green fluorescent protein (*GFP*) gene by recombination with a downstream gene fragment that restores the expression of GFP (Fig. 2f and Supplementary Fig. 4b). Moreover, cellular survival with DNA break and replication stress-inducing agents including cisplatin and PARP inhibitor is not

markedly different in cells containing mutant or wild-type RAD51C (Fig. 2h, i). Taken together, these data suggest that recurrent patient variants that cluster to two distinct CX3 interfaces in three-dimensions show a functional defect despite HDR proficiency. These identified functional interfaces and their implied mutational defects now enable targeted testing of additional cancer patient variants for a comprehensive understanding.

## RAD51C-XRCC3 interaction
RAD51 paralog complex formation requires ATP binding to control protein stability[6,11], which has been used as one measure of functional defects in patient variant classification[3]. In unchallenged cells, RAD51C protein stability is similar in the two mutant HAP1 cells, and XRCC3 protein levels are comparable to wild-type cells (Fig. 3a).

A126 is located on the P-loop, an important ATPase motif that binds the ATP phosphate (Fig. 3b). Despite being consecutive RAD51C residues, the G125V and A126T mutations have different effects on HDR. The structure suggests that the pathogenic G125V mutation will have substantial impact on CX3. Modeling the change from the small and flexible glycine to the bulkier hydrophobic valine, shows insufficient space and a clash with V129, another P-loop residue. This is expected to impact ATP binding and disrupt the CX3 CTD interface. Consistent with these structural implications, yeast two-hybrid data

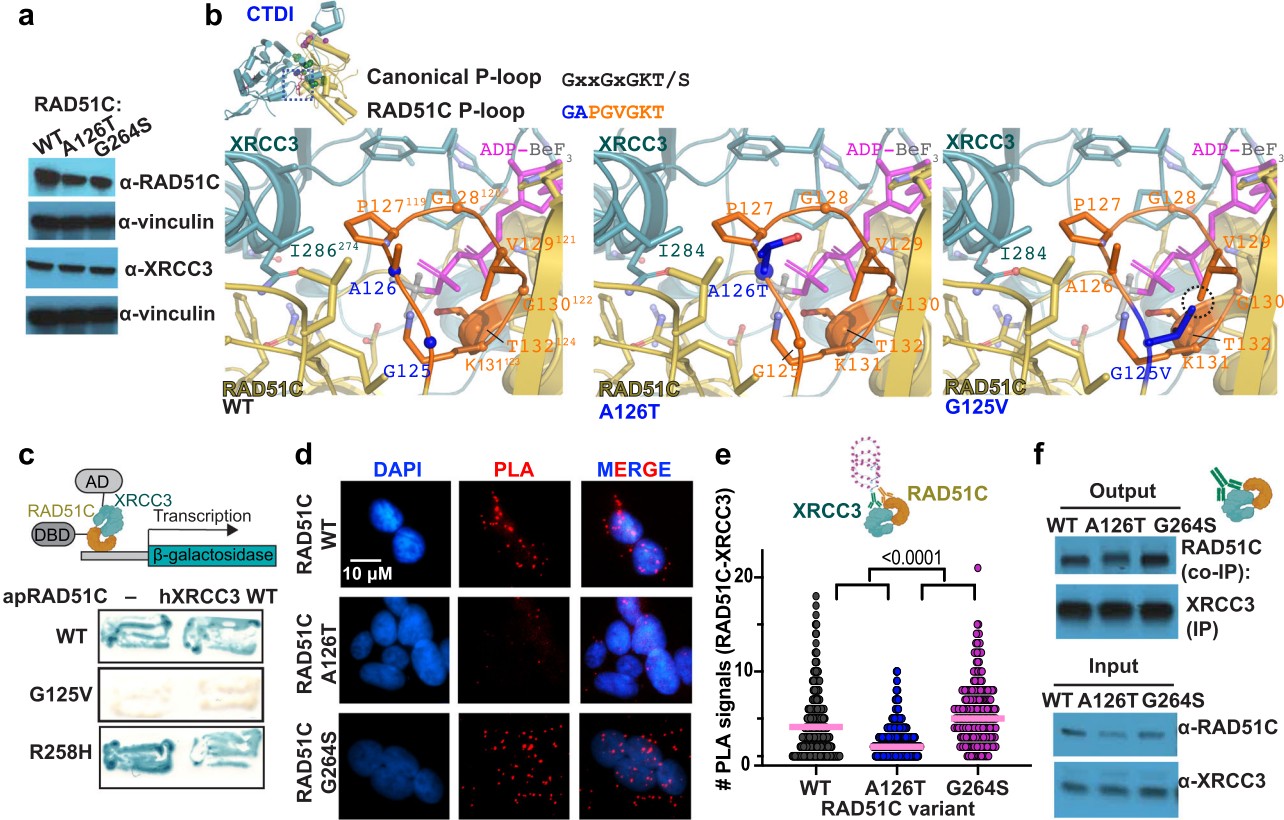

**Fig. 3 | CX3 CTDI mediates RAD51C-XRCC3 paralog binding. a** Western blot of HAP1 cells and variant RAD51C as indicated against RAD51C, XRCC3, or Vinculin as loading control. Representative image from 3 independent biological experiments. **b** Zoomed view of the structural environment of wild-type (WT) CX3 A126 and G125 (left) plus models of the RAD51C A126T (middle) and G125V (right) mutations, with mutated residues in blue and major clashes highlighted by a dashed circle. Top left, ATPase P-loop (orange with blue denoting variant positions) with the canonical and conserved RAD51C sequences. **c** Yeast two-hybrid assay testing interactions between wild-type (WT), G125V and R258H *A. pompejana* RAD51C (apRAD51C) and human XRCC3 (hXRCC3). Top, graphical sketch of two-hybrid assay adapted from http://BioRender.com. **d** Representative images of RAD51C-XRCC3 proximity

ligation assay (PLA, red) in human HAP1 cells containing indicated variant RAD51C. DAPI denotes nucleus. **e** Quantification of RAD51C-XRCC3 PLA signals with hydroxyurea (200 μM) from (**d**), pink bar denotes median. *P* values (<0.0001 between all variants) were calculated using an unpaired two-sided Student T-test, n(WT) = 494, n(A126T) = 601, n(G264S) = 242, derived from 4 independent biological experiments, top, graphical schematic of a PLA reaction created with http://BioRender.com. **f** Immunoprecipitation (IP) of XRCC3 from HAP1 cell extracts expressing wild-type (WT), A126T or G264S RAD51C and Western blot against RAD51C Co-IP. A representative immunoblot from 3 independent experiments is shown. Schematic created with http://BioRender.com.

shows that apRAD51C with the equivalent G125V mutation has a strong interaction defect with human XRCC3 (Fig. 3c), supporting and extending data from recent reports[3].

Besides playing an important role in phosphate binding, in our structure the RAD51C P-loop is adjacent to the XRCC3 CTD interface (Fig. 3b). While we lack yeast two-hybrid data on RAD51C A126T, modeling the A126T mutation shows that there is sufficient space within the structure to accommodate the larger threonine residue, supporting available results that the A126T mutation is less disruptive than G125V[1,3]. However, this mutation is at a critical intersection of XRCC3 and nucleotide binding. Together with our data showing proliferation defects this suggests that the change from alanine to a more rigid Cβ branched threonine impacts CX3 functions due to conformational restrictions. Irradiation of cells has been shown to impact protein stability for XRCC3 but not other paralogs[11], and CX3 has later HDR function after stress[6], suggesting particular importance for these proteins and thus associated protein stability after DNA damage. In addition to HDR roles, RAD51C has roles during replication stress[14–16,34,35]. We therefore tested CX3 paralog interactions in the presence of DNA replication stalling agents. With hydroxyurea, CX3 interactions increase in cells with wild-type RAD51C, but significantly decrease in cells with RAD51C A126T, as measured by proximity ligation assays or by immunoprecipitation (Fig. 3d–f, Supplementary Fig. 5a–c). These results suggest a functional impact of the A126T mutation with replication stalling.

Despite having a significant impact on tumor proliferation, there is robust binding of the G264S variant to XRCC3 slightly exceeding that of wild-type RAD51C (Fig. 3d–f). The data suggest that despite hydrogen bonding and hydrophobic interactions at the NTDI, this region may not affect paralog complex stability per se. While we lack yeast two-hybrid data on G264S, this result is supported by two-hybrid data of the CX3-NTDI-variant R258H that remains proficient for XRCC3 interactions (Fig. 3c, ref. 3). The collective data confirms that CTDI is important for proficient RAD51C-XRCC3 interactions during replication stress, while the NTDI mutation opposes destabilization.

## RAD51C-DNA interaction

The NTDI mutations, with RAD51C R258H located at the beginning of the G264S containing α-helix, point towards a disordered loop in our structure (Fig. 4a). This disordered loop corresponds to the L2-loop in RAD51 that contributes to and becomes ordered upon DNA binding, implying possible impact of this region to DNA binding. We tested single-stranded DNA (ssDNA) binding of purified apCX3 and apRAD51C proteins. Protein-gel-shift analysis confirmed the ability of apCX3 to bind to ssDNA (Supplementary Fig. 6a). Using fluorescence polarization to more rigorously measure DNA binding, the data shows that while wild-type apRAD51C binds ssDNA with an affinity of ~1.25 µM, the RAD51C R258H mutation dramatically reduces DNA binding affinity ~30 fold (~36.5 µM, Fig. 4b).

To further assess CX3-DNA-interaction impacts, we superposed RAD51-DNA and CX3 structures to build a model for how CX3 binds DNA (Fig. 4c). Consistent with the biochemical data showing reduced DNA binding for the R258H mutation, the R258 residue is in close proximity (~4 Å) to the DNA phosphate backbone (Fig. 4c, d). There is also a cluster of conserved hydrophobic residues that stitch this region together and may orient residues for DNA binding. Modeling the R258H mutation suggests that it may disrupt the hydrophobic cluster of residues in this region to impact alignment of DNA binding residues. Furthermore, the shorter length of the histidine versus arginine may hamper it from directly contacting DNA.

G264 maps halfway along the same helix as R258 and is proximal to the XRCC3 NTD interface. While a serine at the 264 position is accommodated without structure clashes at that site, loss of the flexible glycine within this helix plus the bulkier serine sidechain can feasibly impact the functions mediated by this CX3 region (Fig. 4c). We

therefore tested this prediction in cells using the in situ protein interaction with nascent DNA replication forks (SIRF) assay, which measures protein binding to nascent DNA replication forks[36] (Fig. 4e–g). With this assay, nascent DNA is marked by EdU incorporation and subsequent biotinylation. Proximity ligation assay using antibodies against the protein of interest, in this case RAD51C, and against biotin results in a productive RAD51C-SIRF signal only if the protein is within 40 nm of the nascent biotinylated DNA (Supplementary Fig. 6b). In unstressed cells, RAD51C binds weakly to nascently replicated DNA (Supplementary Fig. 6c, d). Replication stalling with hydroxyurea significantly increased RAD51C-nascent DNA binding (Supplementary Fig. 6e), with RAD51C A126T exceeding the DNA binding of wild-type RAD51C (Fig. 4e–g). In contrast, RAD51C G264S-SIRF signals remain low, revealing a significant defect of this variant in associating with stalled forks. These results are not due to differences in the amounts of nascently-labeled DNA available for RAD51C binding since the RAD51C-SIRF signals are normalized to EdU signals. The combined data delineates roles for NTDI in RAD51C-DNA assembly and CTDI in RAD51C-XRCC3 assembly.

## RAD51 stability at replication forks

RAD51C has implied roles in RAD51 loading and filament remodeling leading to stabilization[6,37], which are functions that resemble those of BRCA2. To investigate structural implications for CX3 interactions with RAD51, we first built models by docking our apCX3 structure onto available human RAD51 trimer structures, and supported this assembly with AlphaFold2 prediction of hCX3-RAD51 complex structures (Fig. 5a, b). Our CX3 structure supports and extends observations from RAD51 that a key mechanism for inter-protein interactions in the RAD51 and paralog protein family is the binding of the polymerization motif of one subunit to the CTD of the neighboring subunit, creating a polar dimer. In RAD51, a phenylalanine (F86) in the polymerization motif of one subunit docks into a hydrophobic pocket of the adjacent RAD51[38]. Importantly, we found that CX3 has a unique polymerization motif and related docking sites compared to RAD51.

The polymerization motif of RAD51C does not contain a suitable phenylalanine that could dock into the pocket of an adjacent RAD51, creating an energetically unfavorable state that make these interactions unlikely. Furthermore, RAD51C lacks a CTD pocket suitable for F86 binding, and instead has a phenylalanine that clashes with F86 in models where apRAD51C is superposed with the human RAD51 CTD (Fig. 5b). In contrast, XRCC3 contains a pocket in its CTD, at a position structurally conserved with the RAD51 pocket, which is suitable for F86-mediated RAD51 polymerization motif interaction (Fig. 5b). This supports and extends previous results from yeast two- and three-hybrid assays showing that XRCC3, but not RAD51C, robustly interacts with RAD51[10]. To further support our CX3 structure and structural modeling of the CX3-RAD51 complex, our yeast-two hybrid results show that mutation of the XRCC3 polymerization motif A65E blocks interaction with RAD51C but not RAD51 (Fig. 5c). In addition, mutations of the RAD51 polymerization motif F86, but not CTD hydrophobic pocket, disrupts RAD51-XRCC3 interaction (Fig. 5d), supporting a one-directional binding of RAD51-XRCC3-RAD51C. Given this polarity, the combined data reveal that CX3 is capping a RAD51 filament at the 5′ end. Notably, RAD51-DNA filaments can only assemble or disassemble and dissociate from the 3′ or 5′ DNA end but not from within the filament. Consequently, our structure and the yeast two-hybrid assays supporting CX3 capping of the RAD51 filament at the 5′ end together imply that CX3 blocks filament growth and disassembly on the 5′ end. So this must occur on the 3′ end, thereby regulating the efficiency of functional RAD51 filament growth and stability.

To further test RAD51 stabilization in cells, we utilized RAD51-SIRF to measure RAD51 kinetics at stalled forks as an indirect readout for RAD51-DNA stability (Fig. 5e, f). The expected outcome for a RAD51C mutation affecting RAD51 filament stability is for RAD51 to disassemble

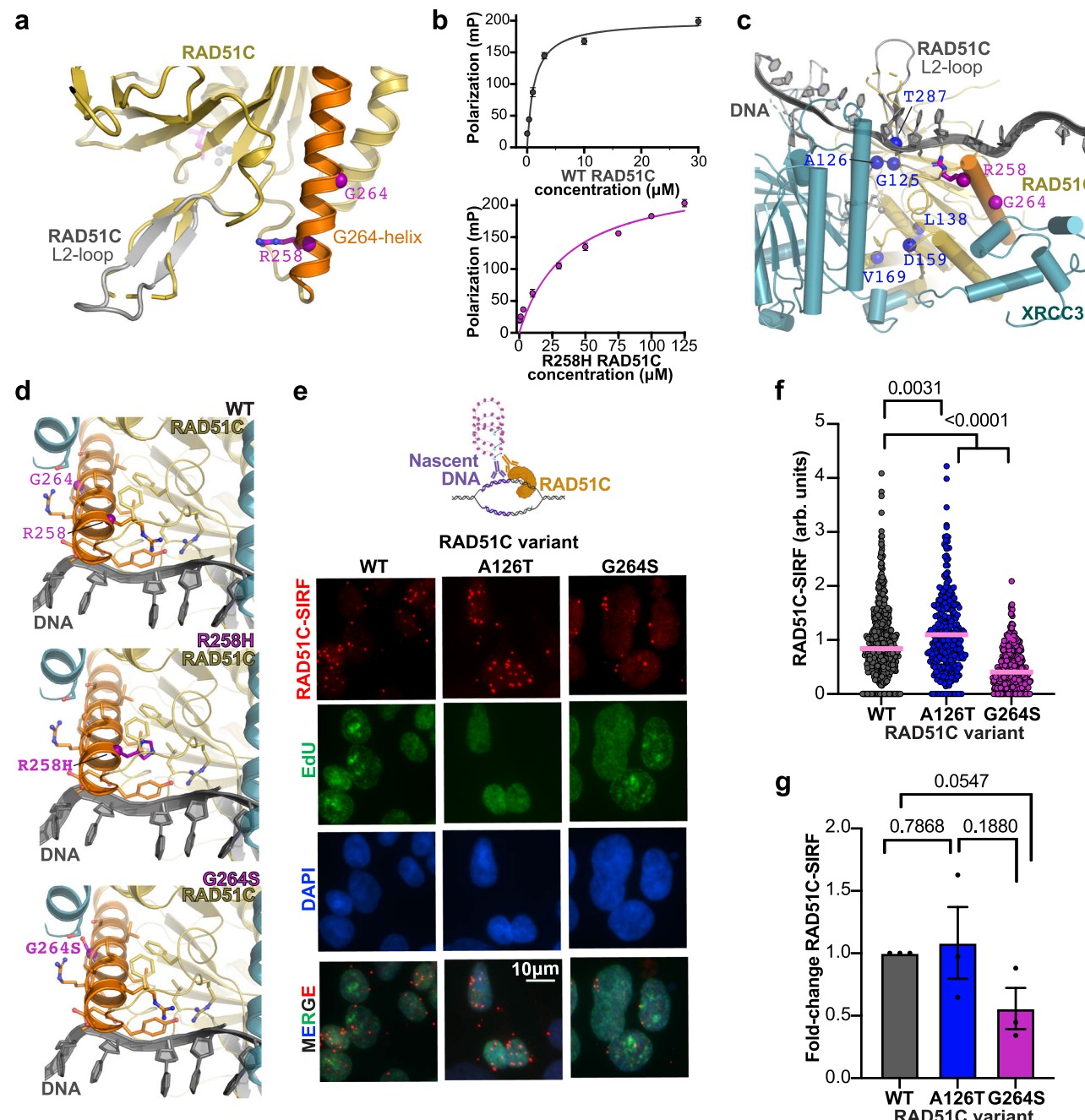

**Fig. 4 | CX3 NTDI mediates RAD51C DNA binding. a** Mapping of the R258H location onto apRAD51C-CTD shows positioning toward a disordered loop that corresponds to the DNA-binding RAD51 L2-loop. The α-helix (orange) containing R258 and G264 (purple) with a model of the RAD51C L2-loop, which closely matches the RAD51 L2-loop seen in DNA bound complexes, is overlaid in gray. **b** Fluorescence polarization assays measuring apRAD51C wild-type (top) and R258H (bottom) binding to single-stranded DNA (ssDNA). For each concentration mean polarization values from $n = 3$ technical replicates and standard deviation error bars are shown. **c** A model of ssDNA (gray) binding to the CX3 complex based on superposition of RAD51-ssDNA structures on the CX3 dimer. RAD51C variants are mapped as for Fig. 2. **d** Zoomed view of the structural environment of CX3 with RAD51C wild-type (WT) with R258 and G264 (top) plus models of the RAD51C R258H (middle) and G264S (bottom) mutations in relation to the modeled ssDNA path. The helix containing R258 and G264 is colored orange with residues within 4.5 Å shown (sticks). **e** Representative images of RAD51C-SIRF assay with 200 μM

hydroxyurea in human HAP1 cells containing indicated variant RAD51C, which produces a red fluorescent signal if RAD51C and EdU labeled biotinylated nascent DNA are in close proximity (<40 nm). Cells were co-clicked with Alexa-Fluor 488 azide and biotin azide to enable simultaneous visualization of newly synthesized DNA (EdU, green) and SIRF signals (RAD51C-SIRF, red), DAPI denotes nucleus. SIRF signals are normalized to total EdU-azide 488 signal to account for potential difference in the amount of nascent DNA available to SIRF signal production (arbitrary unit, a.u.). Top, graphical schematic of a RAD51C-SIRF reaction created with http://BioRender.com. **f** Quantification of RAD51C SIRF signals from (**e**), pink bar denotes median. n(WT) = 360, n(126T) = 341, n(G264S) = 415, *P* values (<0.0001 between G264S and other variants, 0.0031 between WT and A126T) were calculated between each comparison using the two-sided Mann–Whitney test. **g** Fold-change of average RAD51C-SIRF signals in HAP1 cells with indicated RAD51C variants. Data are presented as the mean +/− SEM. *P* values were calculated using an unpaired Student two-sided T-test, *n* = 3 independent biological experiments.

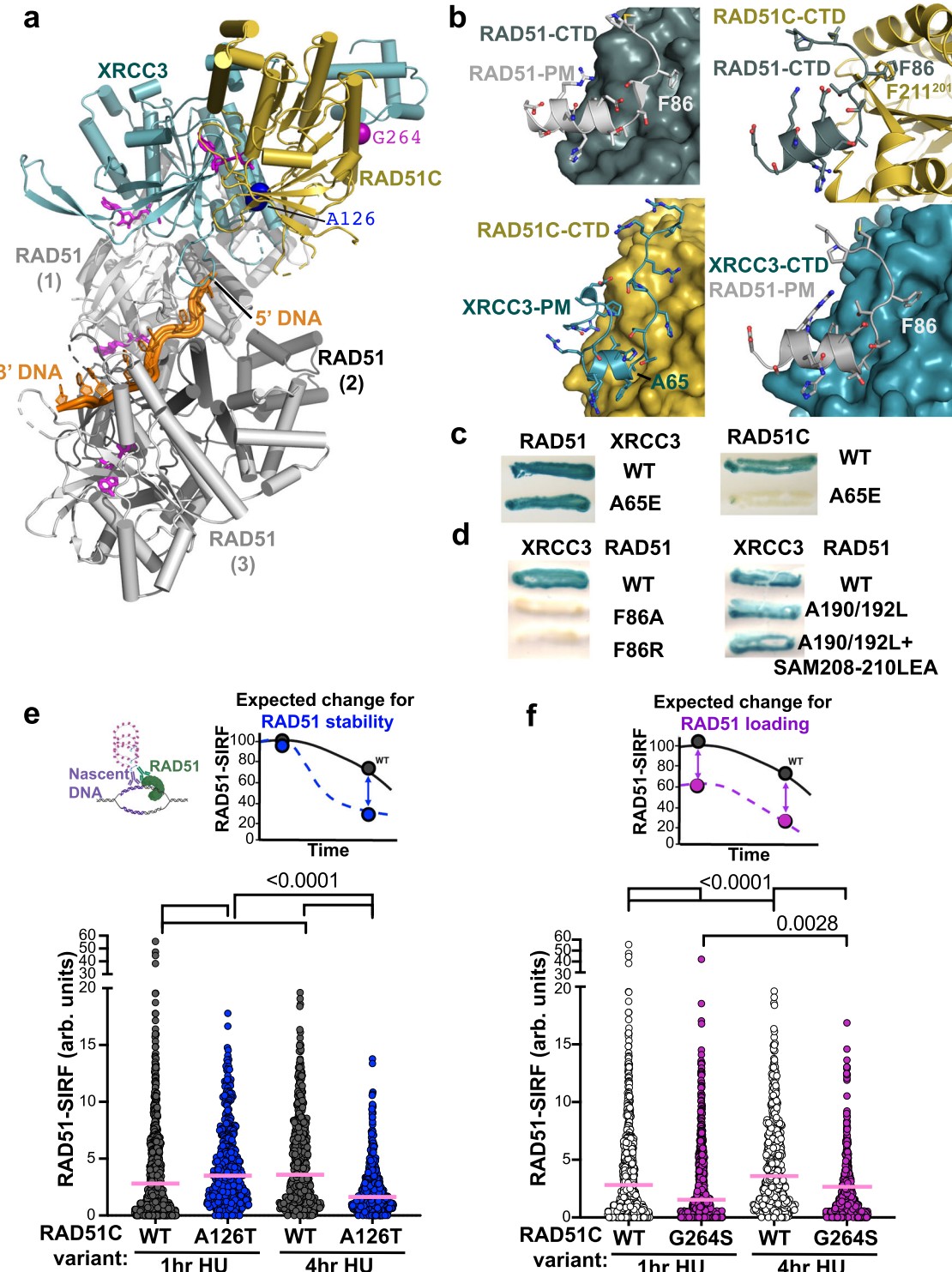

**Fig. 5 | CX3 mediates RAD51 filament loading and stability from structural and cellular analyses. a** Structural model showing how CX3 capping at the 5′-end of RAD51 filaments based on superposition of RAD51-ssDNA filament structures (PDB: 7EJC) with our CX3 structure. **b** Comparison of the canonical RAD51-RAD51 interaction mediated by the F86 polymerization motif (PM) (top left) to the XRCC3 PM interaction with RAD51C (bottom left) and predicted interactions of the RAD51 PMf with RAD51C (top right) and XRCC3 (bottom right) CTDs. **c** Yeast two-hybrid assay testing interactions between wild-type (WT) RAD51 and RAD51C with XRCC3 WT and PM A65E. **d** Yeast two-hybrid assay testing interactions between WT XRCC3 with WT RAD51 and indicated PM F86 (left) and CTD pocket (right) mutations. **e** Quantification of RAD51-SIRF signals in wild-type (WT) and RAD51C A126T HAP1 cells 1 h and 4 h after replication stalling with hydroxyurea, pink bar denotes

median. Top left, graphical schematic of a RAD51-SIRF reaction created with http://BioRender.com, top right, expected outcome for RAD51 filament stabilization defects (blue). n(WT, 1 h) = 342, n(A126T, 1 h) = 463, n(WT, 4 h) = 347, n(A126T, 4 h) = 417, from 3 independent biological experiments. **f** Quantification of RAD51-SIRF signals in RAD51C G264S HAP1 cells 1 h and 4 h after replication stalling with hydroxyurea, pink bar denotes median. Data from wild-type (WT) HAP1 cells is replotted from (**e**), top right, expected outcome for RAD51 filament loading defects (purple). n(G264S, 1 h) = 701, n(G264S, 4 h) = 723, from 4 independent biological experiments. *P* values for all RAD51 SIRFs (<0.0001 for all comparisons except 0.0028 for G264S comparison at 1 h and 4 h post HU) were calculated between each comparison using the two-sided Mann–Whitney test.

faster from the stalled forks compared to wild-type RAD51C (Fig. 5e, blue sketch on top). With fork stabilization being affected, but not RAD51 loading, decreased RAD51-SIRF signals would be expected at later time points after fork stalling, but not at early time points. Conversely, compromised RAD51 loading onto stalled forks is expected to result in less RAD51-SIRF signals at both early and late time (Fig. 5f, purple sketch on top).

Irrespective of the RAD51C variant present, we find little RAD51-SIRF signal in unchallenged cells, which is vastly increased with replication stalling by hydroxyurea (Supplementary Fig. 7a, b). We observe a small increase in RAD51-SIRF signals at early times after replication but a significant decrease with prolonged replication stalling in HAP1 cells containing the CTDI RAD51C A126T variant compared to wild-type RAD51C (Fig. 5e). Cells containing the NTDI RAD51C G264S variant, on the other hand, show less RAD51-SIRF signals at both early and late time points. Together the data is consistent with RAD51C A126T affecting RAD51 filament stability resulting in faster RAD51 disassembly from stalled forks, while the RAD51C G264S mutation directly or indirectly prevents efficient RAD51 assembly at stalled forks.

### Distinct RAD51C replication roles

Diverse replication functions have been ascribed to RAD51C including fork protection, fork restart, and fork remodeling. Fork protection necessitates RAD51 filament stabilization[17] and RAD51C ATP binding, but not hydrolysis[15]. We tested the RAD51C variants for fork protection using DNA fiber analysis with dual labeling of nascent DNA by IdU followed by CldU in the presence of high concentrations of hydroxyurea (HU). Loss of IdU tract length under these conditions is a measure of fork protection defects (Fig. 6a). IdU tracts are significantly shortened in cells containing RAD51C A126T compared to those of wild-type RAD51C cells, with RAD51C G264S exhibiting a much milder effect (Fig. 6a).

Requiring both ATP binding and ATP hydrolysis, RAD51C promotes replication restart[15]. Testing the efficiency of replication resumption by measuring the CldU tract length after replication with hydroxyurea reveals a strong restart defect in cells containing RAD51C G264S mutations compared to wild-type and RAD51C A126T mutations (Fig. 6b). These data with the G264S NTDI mutant support and extend previous reports in hamster cells, where overexpression of the NTDI mutant RAD51C R258H causes a defect in fork restart but not fork protection[15], phenocopying RAD51C G264S. Moreover, primary mouse adult fibroblasts containing a hypomorphic small deletion within the NTDI sandwiched between the arginine and the glycine equivalent to hR258 and hG264 also causes a strong replication restart defect with only mild effects on fork protection[5]. The combined data support the importance of the NTDI for efficient DNA binding and RAD51C-mediated replication restart.

To further test CTDI replication functions, we analyzed HAP1 cells endogenously expressing RAD51C T287A (Fig. 6c–e, Supplementary Fig. 8a, b). While T287 is close to NTDI residues in the primary sequence, the three-dimensional structure places it as part of the CTDI. T287 maps to the edge of the CTDI on the base of the L2-loop and near where DNA is predicted to contact the CTD domains of both RAD51C and XRCC3 (Fig. 6c). Interestingly, this positions T287 between the nearby RAD51C P-loop G125 and A126 residues and the NTDI R258 residue that is on the opposite side of the projected DNA path. Consistent with predominantly CTDI functions, we find that RAD51C T287A causes a strong fork protection but no significant restart defect (Fig. 6d, e), further supporting the suggestion of a more pronounced role of the CTDI in fork protection compared to restart.

RAD51 and paralog functions at stalled replication forks also involve fork reversal. While currently electron microscopy is the only technique to directly visualize reversed replication forks, replication slowing with the topoisomerase II-DNA crosslinking agent camptothecin (CTP), which can be measured by a decreased CldU/IdU tract ratio, can be used as a surrogate marker for fork reversal[13,39,40]. The data reveals that despite being replication restart defective, cells containing RAD51C G264S slow replication slightly more effectively than cells containing wild-type RAD51C, suggesting fork reversal proficiency (Fig. 6f). CldU/IdU tract ratio in cells with RAD51C A126T on the other hand is remarkably increased after CTP treatment compared to either wild type or RAD51C G264 cells, suggesting a significant loss of reversed forks in these cells. These collective data uncover distinct replication stress response defects of RAD51C patient variants previously thought to be functionally proficient based on the data in hand.

## Discussion

Together, combined structural and replication analyses inform on and define functionally distinct regions for RAD51C (Fig. 6g). Notably, these structures from X-ray crystallography and scattering define a conserved paralog structural platform for RAD51 binding (ATP-bound XRCC3) including its intact polymerization motif linker and N-terminal domain connections to its paralog partner (RAD51C) with their replication functions. Prior to these structural analyses, the only recognizable motif CX3 paralogs possessed were Walker motifs[11] (Fig. 1a). The CX3 co-crystal structure provides a fundamental structural reason for how the RAD51 polymerization motif can bind to XRCC3 but not RAD51C CTD. Moreover, the RAD51 polymerization motif is void of a phenylalanine critical for RAD51 CTD core binding. We propose that these unique structural features of RAD51C force a 5′ capping of the RAD51 filament by CX3 that can stabilize RAD51 filaments and dictate filament growth and disassembly at the 3′ of the DNA, which is the functionally active end for DNA polymerizations. RAD51 filament capping in an ATP-dependent manner is suitable to induce a conformational transition for extended and open ssDNA. Our observations support and extend implications from *C. elegans* RFS-1/RIP-1 RAD51 paralogs in paralog-mediated filament stabilization, which requires ATP binding but not hydrolysis for filament remodeling and stabilization[37]. Unidirectional filament growth and ATP-mediated stabilization can inform the results seen in mutant RAD51C A126T cells with increased RAD51-SIRF signals reflecting inefficient RAD51 filament capping early but decreased RAD51-SIRF signals later on due to filament instability. Stabilized RAD51 filaments are required during fork protection, an HDR-independent genome stability promoting mechanism[17]. RAD51C-mediated fork protection requires ATP binding, but not hydrolysis[15]. This implies that the A126T mutation may affect ATP binding without strongly affecting ATP hydrolysis.

Guided by cancer patient variants and the CX3 structure, we define a CTDI region on RAD51C important for fork protection and ATP binding (Fig. 6g). The more open RAD51 filament from CX3-mediated capping may also be more amicable for the DNA pairing required for both HDR and fork reversal. The RAD51C A126T mutation causes fork protection defects and is also defective in replication fork reversal-associated slowing of replication forks with CTP, reiterating these concepts. While this seemingly contradicts current models of fork protection occurring at reversed replication forks[41], in which we would expect more (and not less) fork protection with fewer reversed forks, it is consistent with this biophysical assessment. Notably, the combined insights from this study define added roles to the RAD51C CTDI in addition to the previously inferred paralog interaction interface. The exact mechanism of how RAD51 paralogs and RAD51 contribute to fork reversal is not understood and merits further investigation in particular in light of emerging fork protection functions at gaps behind the fork[42], which are mutually exclusive with reversed forks. Of note, an undefined region within RAD51C residues 1–126 was implicated in Holliday junction resolution, a DNA structure resembling reversed forks[12]. Moreover, RAD51C and XRCC3 but not XRCC2 have been associated with longer replication tracts in hamster, chicken, and human U2OS cells after CTP stalling[16,40,43], implying fewer reversed forks with XRCC3 deficiency. Interestingly, XRCC3 fork slowing after

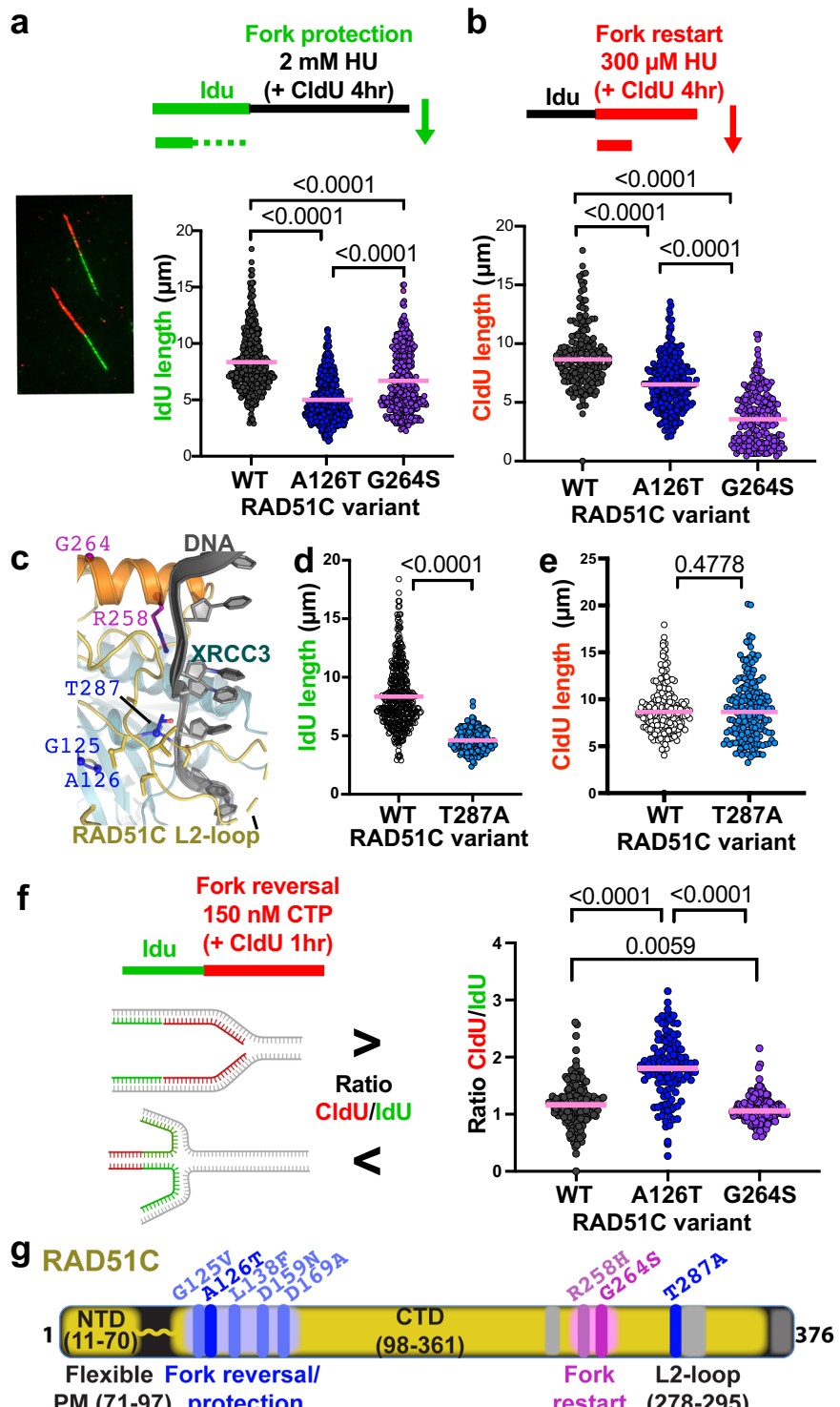

replication stalling is epistatic with PARP1[43], a protein that does not promote but rather inhibits premature resolution of reversed forks[44]. Consistent with a requirement for CX3 downstream of BCDX2[6] and the observed apparent increase in slowed forks with RAD51C G264S mutations, we suggest that combined observations imply CX3 roles during resolution of reversed forks rather than in their creation, which will be important to delineate in future studies.

RAD51 and paralog defects are associated with both cancer and chemotherapeutic sensitivity, including to PARP inhibitors (PARPi). The molecular causes of PARPi sensitivity are in debate and may involve promoting fork collapse, fork protection, fork restart, DNA

gaps, HDR, replication slowing, or these functions combined. Importantly, in the clinical setting CX3, but not BCDX2, associates with PARPi sensitivity[7], an observation that suggests a path forward to further enable functional understanding of RAD51 paralogs.

RAD51C-mediated replication fork restart requires ATP hydrolysis[15], a reaction that in RAD51 is triggered by DNA binding[45]. We define a previously unappreciated DNA binding region of RAD51C at the NTDI required for replication restart (Fig. 6f). CX3 ATP hydrolysis likely promotes disassembly of the 5'cap. Complex disassembly requiring DNA binding is supported by NTDI/DNA binding mutant RAD51C G264S showing slightly increased XRCC3 binding compared

**Fig. 6 | Distinct replication stress reactions by CX3 CTDI and NTDI. a** Dot-blots of nascent IdU replication tracts lengths in human HAP1 cells containing indicated variant RAD51C. left, Representative image of single-molecule DNA fiber tracts labeled with IdU (green) and CldU (red). top, schematic of DNA fiber labeling with IdU, followed by CldU with high concentrations of hydroxyurea (HU) to measure fork protection. $n = 4$ independent biological experiments. n(WT) = 411, n(A126T) = 455, n(G264S) = 379, derived from 4 independent biological experiments. **b** Dot-blots of nascent CdU replication tracts lengths in human HAP1 cells containing indicated variant RAD51C. top, schematic of DNA fiber labeling with IdU, followed by CldU with low concentrations of hydroxyurea (HU) to measure replication fork restart. n(WT) = 185, n(A126T) = 275, n(G264S) = 211, derived from 3 independent biological experiments. **c** Zoomed view, using the CX3-DNA model from Fig. 4c, of the structural environment of RAD51C T287A with DNA, P-loop variants, G264-helix variants and residues within 4.5 Å of T287 (sticks) highlighted. **d** Dot-blots of nascent IdU replication tracts lengths followed by CldU with 2 mM of

HU as in panel (**a**), in human HAP1 RAD51C T287A cells, results for wild-type HAP1 are replotted. n(T287A) = 170, derived from 2 independent biological experiments. **e** Dot-blots of nascent CldU replication tracts lengths in the presence of 300 μM HU as in panel (**b**), in human HAP1 RAD51C T287A cells, results for wild-type HAP1 are replotted. n(T287A) = 188, derived from 2 independent biological experiments. **f** Dot-blots of ratio of nascent CdU divided by IdU replication tracts lengths in human HAP1 cells containing indicated variant RAD51C. top, schematic created with http://BioRender.com of DNA fiber labeling with IdU, followed by CldU with low concentrations of camptothecin (CPT) to measure replication fork slowing during CPT indicative of replication fork reversal. n(WT) = 162, n(A126T) = 124, n(G264S) = 142, derived from 2 independent biological experiments. *P* values (shown above brackets in the figure panels) for all DNA fiber analysis and between each comparison were calculated using the two-sided Mann–Whitney test. **g** Schematic figure of RAD51C protein highlighting regions responsible for fork protection in blue and fork restart in pink.

to wild-type RAD51C, which functionally connects CTDI and NTDI. Inability to resolve reversed forks moreover would result in a restart defect and accumulation of reversed forks, as we observed with RAD51C G264S mutations. Moreover, while RAD51C T287 is not part of the NTDI, in our model it is near the DNA on the opposite side to the NTDI R258 DNA binding residue, further suggesting that CTDI and NTDI regions are acting in concert within the CX3 complex. Thus, some of the phenotypes of other variants may not be as clear cut as found for the exemplary variants examined here. Notably, despite showing a defect, this RAD51C G264S remains proficient for paralog protein complex assembly, a parameter currently used for functional patient VUS testing in addition to HDR activity.

In vitro and in situ studies as performed here and elsewhere provide understanding of protein function and how mutations diminish them. Our data shows that the examined patient-associated paralog mutations are damaging and cause replication-associated genome instability, which is a hallmark of cancer. A damaging mutation may not be sufficient as a cancer driver mutation. However, following the Knudson two-hit model for cancer development[46], a mutational defect may drive cancer in the context of a second mutation. This is exemplified by recent preclinical studies in mice with hypomorphic *Rad51c* NTDI mutation that promote a restart defect[5] consistent with our studies here. On its own, the mutation appears benign. However, in the context of a second also seemingly benign *Brca2* mutation, the added *Rad51c* NTDI mutation strongly drives cancer in these mice and confers cancer therapy sensitivity[5]. This observation may explain why some mutations can exist in relatively high frequency in the general population yet be associated with cancer development when in an environment with additional tumor suppressors, resulting in discrepant interpretation based on population studies alone[23–30]. Given that cancer requires more than one gene defect, understanding of such secondary driver mutations may prove critically important in the context of advising patients and treatment strategies.

Together, our findings reveal an emerging importance of CX3 in particular during DNA replication stress. Consequently, the data indicate that cancer variant testing could be expanded to include DNA replication and related functions for a comprehensive understanding of the potential defects of damaging mutations. Our results and insights on CX3 structure and replication biology advance knowledge of the multiverse of RAD51 paralog functions contributing to BRCA-ness that may aid an actionable understanding of cancer biology and therapy including PARPi sensitivity.

## Methods

### Identification of Xrcc3 and Rad51C human homologs in *Alvinella pompejana*

In 2009, during several expeditions on the deep-see submersible Alvin, we collected adult annelid specimens around super-heated (-150 °C)

metal ion- and sulfide-rich hydrothermal venting chimneys along the East Pacific Rise, between the Pacific and North American Plates (9°N, 50/104°W17) (19063897) at -−2500 m. These samples were used to generate an expressed sequence tag library, which was later merged with two additional EST libraries to generate a composite set of predicted *A. pompejana* genes (23324155) (Alvi_3est)[47]. In addition, we generated ~43 gigabytes (Gb) of short whole genome sequencing reads using an Illumina platform and used SOAPdenovo (https://www.animalgenome.org/bioinfo/resources/manuals/SOAP.html) to assemble reads into larger scaffolds (contig N50, 3657 bp; scaffold N50, 22.13 kb) (33300026). These partial assemblies were insufficient to generate a draft genome of *A. pompejana* genome, but sufficient to identify several hundred genes of interest. To this end, we used a subset of 412 bona-fide codon sequences from Alvi_3est to train Augustus (http://bioinf.uni-greifswald.de/augustus/) on intron-exon boundaries and performed an ab initio prediction of genes, both on the scaffold library and Alvi_3est. We then used human RAD51C (UniProt O43502) and protein sequences as baits in blastP searches; the top alignments were g45829.t1 (bit score 392; E value 7e−136, with start_codon, stop_codon, 5 introns) for RAD51C and g58419.t1 (bit score 281; E value 5e−93, with start_codon, stop_codon, 6 introns) for XRCC3.

### Cloning, expression, and purification

For structural work primers were designed to clone the RAD51C and XRCC3 expression constructs indicated in Supplementary 1a, which include apRAD51C NTD and CTD subdomains, the core apCX3 complex and hCX3. ApRAD51C NTD and CTD were amplified from *A. pompejana* cDNA libraries available in the Tainer lab. ApRAD51C CTD was cloned into the NdeI and XhoI restriction sites of pET24 to include a C-terminal His-tag, and RAD51C NTD was cloned into the NdeI and BamHI of a modified pET24b vector to include a TEV-cleavable His-MBP tag at the N-terminus. The R258H mutation was introduced into the ApRAD51C CTD construct using site-directed mutagenesis. Synthesized and codon optimized templates were used to amplify apRAD51C and apXRCC3 to generate the apCX3 core expression construct in pRsf-Duet1, with apRAD51 (84–362) and full-length apXRCC3 cloned into the NdeI and AvrII and NcoI/NotI restriction sites, respectively. hRAD51C and hXRCC3 were cloned into the 11B and 11C MacroBac insect cell expression vectors[48], to respectively include TEV-cleavable His- and His-MBP tags. All cloned expression vectors were sequence verified. apRAD51C and apCX3 proteins were expressed in Rosetta(DES) *E. coli* cells using Terrific (apRAD51C-NTD and CTD) or Turbobroth (apCX3). Cells were grown to OD 0.6–1 at 37 °C, induced with 200 μM IPTG and expressed at 15–16 °C O/N for 15 h. The hCX3 coexpression vector was transformed into DH10Bac *E. coli* cells to generate bacmids that were used to infect SF9 insect cells for protein expression following the Bac-to-Bac protocol (Thermo Fisher Scientific). Insect cells were grown to amplify the virus three times and protein expression occurred over 72 h.

Expression cultures were harvested, resuspended in 20 mM Tris 8.0, 0.5 M NaCl, 20 mM imidazole, 5 mM TCEP (apCX3-core complex and apRAD51C-NTD) or 25 mM HEPES 8.0, 0.4 M NaCl, 20 mM imidazole, 2.5 mM ATP, 2.5 mM $MgCl_2$, 0.2 mM $Na_3VO_4$ (apRAD51C-CTD and hCX3). For lysis, resuspension buffer also contained complete EDTA-free protease inhibitor, DNaseI and lysozyme. Cells were lysed by an Avestin Emulsiflex C5 (apCX3) or sonication (apRAD51C-NTD, -CTD and hCX3). Lysed cultures were centrifuged at 25,000–50,000 × $g$ for 1 h at 4 C. Clarified supernatant was applied to NiNTA beads in the same resuspension buffer (minus protease inhibitors) and contaminants removed with extensive washing with 30–100 mM Imidazole. Bounded protein was eluted in the resuspension buffer plus 300–400 mM imidazole. For tag removal (ApRAD51C NTD, hCX3) protein was dialyzed overnight with His-TEV protease (purified in house) into resuspension buffer followed by application to a HisTrap column (Cytiva) to separate cleaved protein, which flowed through the column or was removed with 5–15 mM imidazole washes, from His-tagged contaminants that bound the column. Size exclusion chromatography (Cytiva Superdex 200, Superdex 75, or Sephacryl 200 depending on the size of the protein) was used as a final purification step for all proteins. Protein was concentrated in spin concentrators with an appropriate molecular weight cutoff (Cytiva) to 5–25 mg/mL. ApRAD51C NTD, CTD, and hCX3 proteins were exchanged using Zeba desalting columns (Thermo Fisher Scientific) into 10 mM HEPES pH 8, 100 mM NaCl, 2 mM DTT, with 2.5 mM ATP, 2.5 mM $MgCl_2$ and 0.2 mM $Na_3VO_4$ also added to stabilize ApRAD51C CTD and hCX3. apCX3 complex was gel filtered into 10 mM TRIS pH 8, 300 mM KCl, 3 mM TCEP. If needed, protein aliquots were flash-frozen in liquid nitrogen and stored at −80 °C prior to use. hCX3 complex was also purified as above except in the absence of ATP, $MgCl_2$, and vanadate.

## Crystallization and structure determination
All crystallizations used the sitting drop method at 15 °C. apCX3 (25 mg/mL)+ 2.5 mM ADP-BeF, 5 mM $MgCl_2$, grew with a 1:2 protein:crystallization reagent (o.1 M HEPES pH 7.5, 20% PEG 3350, 2% Tacsimate pH 7.0) ratio and reached maximum growth after 15 days. Crystals were harvested and frozen in liquid nitrogen for data collection. apRAD51C CTD (25 mg/mL) crystallized in 1:1 protein:crystallization reagent (0.1 M Bis-TRIS pH 5.5, 0.3 M Na formate), with best diffraction achieved after additive screen optimization with 4% 1,3-propanediol and 10 days of growth. apRAD51C NTD (13 mg/mL) crystallized in 1:1 protein:crystallization reagent (0.05 mM Zn acetate and 20% PEG 3350) and crystals were observed 3 months after setup. apRAD51C Crystals were harvested, cryoprotected in crystallization buffer plus 15% 2,3-butanediol and frozen in liquid nitrogen for data collection.

All X-ray diffraction data was collected at beamline 8.3.1 at the Advanced Light Source, Lawrence Berkeley National Laboratory. apCX3 data, collected in a liquid nitrogen stream at a wavelength of 1.11583 Å, was processed with XDS[49], the structure phased using molecular replacement with RAD51 search models in PHASER[50], with rounds of model building and refinement done in Phenix[51] and COOT[52]. apRAD51C CTD and apRAD51C NTD data, both collected in a liquid nitrogen stream at a wavelength of 1.11587 Å, was processed with Mosflm[53] and SCALA[54]. The apRAD51C CTD structure was phased using molecular replacement in Phaser with a RAD51 search model. The apRAD51C NTD structure was phased in SHELX[55] by single-wavelength anomalous dispersion using anomalous signal from a zinc ion bound to the N-terminus of the protein and partly coordinated by a leftover purification tag residue. apRAD51C CTD and apRAD51C NTD structures were build and refined in COOT and REFMAC[56]. All structural figures in the manuscript were made in the PyMOL Molecular Graphics System, Version 2.5 Schrödinger, LLC. For Fig. 4, to build the model of apCX3 DNA binding, the RAD51-ssDNA bound structure (PDB 7ejc[57]) was superimposed with our CX3 structure and the RAD51 subunits were

removed to highlight a DNA path consistent with how DNA binds RAD51. To highlight L2-loops that are disordered in our experimental structures, we overlaid models of these loops predicted by AlphaFold2[58]. For Fig. 5, the canonical RAD51 polymerization motif is shown based on available RAD51 structures (PDB 7ejc). The models for RAD51-XRCC3 and RAD51C-RAD51 polymerization motif interactions are based on AlphaFold2 predictions. All structure PDB files are available in the protein databank.

## Small-angle X-ray scattering analysis
SAXS data was collected at the Advanced Light Source SIBYLS beamline (12.3.1) at Lawrence Berkeley National Laboratory using an X-ray wavelength of 1.0 Å at 20 °C[59]. Purified hCX3 complex was buffer exchanged into SAXS buffer (10 mM HEPES pH 8, 100 mM NaCl, 2.5 mM ATP, 2.5 mM $MgCl_2$, and 0.1 mM $NaVO_4$) using Zeba spin columns immediately prior to data collection. apCX3 concentrations of 4, 2, and 1 mg/mL were made by diluting stock protein with SAXS buffer immediately before data collection. SAXS data was collected on protein sample and SAXS buffer with 0.2 and 1 s exposures, and scattering curves generated by subtracting buffer from protein samples. This approach was also used to collect SAXS data on hCX3 protein purified in the absence of ATP, vanadate and $MgCl_2$. Scattering curves were analyzed, scaled, and merged in scÅtter (from beamline 12.3.1, advanced light source) to generate SAXS data across the entire scattering spectrum. A model of full-length hCX3 was generated in iTasser using our apCX3 core and apRAD51C NTD structures as a template with a structural model of the connecting linker (hRAD51C 70–94). Molecular dynamics and minimal ensemble search approaches using the hCX3 full-length model, with the regions corresponding to apCX3 and apRAD51C NTD structures fixed as rigid bodies and the linker flexible, were done in BILBOMD[22] to identify the model that best fit the hCX3 SAXS data.

## DNA binding assays
20 μL reactions containing the indicated ApRAD51C-CTD1 concentration of protein (WT or R258H) and 10 nM 5'fluoresceine-labeled ssDNA substrate (poly-20dT) in DNA binding buffer (20 mM HEPES pH 8, 80 mM NaCl, 2.5 mM ATP, 2.5 mM $MgCl_2$, 1 mM DTT, and 0.1 mg/mL BSA). Samples were incubated in the dark for 20 min at 37 °C before fluorescence polarization data was collected in triplicate by measuring fluorescence at 520 λ with 0.2 s exposures using a TECAN Infinite 200 PRO. Data was plotted and standard deviation errors were calculated from triplicate measurements in PRISM, with the curve fit with a Michaelis-Menten non-linear fit. The $K_d$ value for the R258H mutation is only an estimate because the curve did not fully saturate with these conditions.

Electromobility shift DNA binding assays (EMSA) were done using 1% agarose gels in 0.5 X TBE buffer (Life Technologies) equilibrated at 4 °C and pre-run at 30 V for 1 h. 10 μL reactions with poly-21dT (ssDNA with an internal FAM label on the 11th dT) and the indicated ratios of apCX3:DNA were setup in EMSA buffer (20 mM Tris pH 8.0, 150 mM KCl) and incubated in the dark at 4 °C for 15 min. Subsequently, 1 μl of loading buffer (20 mM Tris pH 8.0, 40% glycerol) was added to each sample before running on the agarose gel (30 V at 4 °C) in the dark for 2.5 h before DNA electromobility was visualized by exposure to UV radiation.

## Yeast two-hybrid assays
Previously generated yeast two hybrid vectors used here were human XRCC3, fused to pGal4 activation domain, and RAD51, fused to the pGal4 DNA binding domain[60]. The ApRAD51C yeast two-hybrid vector was made by cloning apRAD51C cDNA into the EcoRI and BamHI sites of the pGBT9 vector, creating a pGal4 DNA binding domain fusion. WT yeast two-hybrid vectors were used as a template for site-directed mutagenesis to make all indicated point mutations. Yeast two-hybrid assays in the Y190 yeast strain were then carried out using standard

approaches for RAD51 paralogs[60]. Protein–protein interactions comparing WT and mutant proteins were qualitatively compared using β-galactosidase X-gal filter assays, with a blue readout indicating an interaction.

## Cell lines, siRNA, and reagents

HAP1 cells (cat# C631, Horizon Genomics GmbH) were grown in Iscove's modified Dulbecco's medium (IMDM) (Life Technologies, 12440061) supplemented with 10% fetal bovine serum (FBS) (Gemini Bio Products, 100-106) and penicillin-streptomycin (100 U/mL) (Gemini Bio Products, 400-109). All cells are grown at 37 °C and 5% $CO_2$. *RAD51C* variant knock-ins were created by Haplogen Genomics GMBH as previously reported[9], whereby guide RNAs used for targeting G264S were TGCAAGGCTGATCATTTGCT, for A126T ATTTGTGGTG-CACCAGGTGT, and for T287A AATCTTTGTTGTCATCTGAT and a ~200 nucleotide ssDNA template was provided for variant knock-in, creating PAM sites with silent modifications. The same clone was used for biological repeat experiments as indicated in the Figure legends. Authentication was performed by STR fingerprint and Sanger sequencing (see Supplemental Fig. 4a). Hydroxyurea (HU, H8627), BrdU, 5-iodo-2′-deoxyuridine (IdU, I7125), 5-chloro-2′-deoxyuridine thymidine (CldU, C6891), and Duolink proximity ligation assay reagents (DUO92001, DUO92005, and DUO92008) were from Sigma-Aldrich; EdU (A10044), biotin azide (B10184), Alexa Fluor 488 azide (A10266) and ProLong Gold (P36934) were from Invitrogen; 4′,6-dia-midino-2-phenylindole (DAPI, 62248) was from Thermo Fisher Scientific; 32% Paraformaldehyde (PFA, 15714) was from Electron Microscopy Science; Camptothecin (S1288) was from Selleck chemicals; For PLA assays XRCC3 SiRNAs (SI00077126, SI00077119, SI00077112) were purchased from Qiagen and RAD51C SiRNA (M-010534-01-0005) was from Dharmacon. SiRNAs were transfected into cells using RNAiMAX (Invitrogen, 13778-150) according to the manufacturer's protocol.

## Antibodies

Antibodies used in SIRF and PLA assays are as follows: mouse anti-biotin (Sigma-Aldrich, BN-34), rabbit anti-biotin (Cell Signaling Technology, D5A7), rabbit anti-XRCC3 (Abnova, PAB24835), mouse anti-RAD51C (Abnova, H00005889-M01), rabbit anti-RAD51C (Abcam, ab72063), and mouse anti-RAD51 (Abcam, ab213). Antibodies used in immunoblotting and immunoprecipitation are mouse anti-RAD51C (Novus, NB100-177) and rabbit anti-XRCC3 (Novus, NB100-165). Antibodies used in DNA fiber assays are anti-IDU (BrdU, Beckton Dickinson, 347580, 1:50) and CldU (BrdU, Abcam, ab6326, 1:100). More detailed antibody information can be found in the Supplementary Table 3. Uncropped and unprocessed scans of Western blots and gels can be found in the Source data file.

## DNA isolation and Sanger-sequencing

Total DNA was isolated using Direct PCR Lysis buffer (Viagen) followed by endpoint PCR. The following primer sequences were used: RAD51C (A126T): Forward: 5′-GTGTACAGCACTGGAACTTCTTGAGC-3′, Reverse: 5′-GCATACATTTATCAAGAAGGGATAATG-3′. RAD51C (G264A): Forward: 5′-AGCACTGGCTGAACAGCTTTG-3′, Reverse: 5′-CTACCGCGCGCTCAACCACAAAGTCCA-3′. PCR products were purified using PureLink PCR Purification Kit (ThermoFisher) and Sanger sequenced and then analyzed using Snapgene software.

## Mouse models and xenografts

Immunodeficient mice (*NOD.Cg-Prkdc"id Jl2rg,mlWjI/SzJ*, Stock 005557, JAX) were housed in a sterile environment and allowed free access to food and water. The animal experiments were approved by the *institutional animal care and use committee* (IACUC) and was described in an *Animal Care* and Use Form (ACUF, protocol nr 00001436-RN01). All procedures and methods

were performed according to the federal and state regulations as well as MD Anderson Cancer Center institutional guidelines and policies for the protection of animals. HAP1 cells were harvested according to standard cultivation conditions and suspended in PBS with 50% Matrigel. HAP1 xenografts were initiated by injecting $1 \times 10^6$ cells subcutaneously in the right and left flank of 8–12-week-old mice. Tumors were measured at least two times a week using calipers when tumor volumes $[(L \times W \times H \times \pi)/6]$ reached a suitable size (500–1000 mm³). If the tumor volume of 1500 mm³ was reached the mouse was euthanized and removed from the experiment group. Mice have not exceeded 20 mm at the largest diameter of tumor size. Animal weights were also monitored. The in vivo experiments were performed at least twice to reach a group size between 18 and 44 tumors.

## Cellular survival assays

For colony formation assay, 75 HAP1 cells were plated in a 12-well plates and treated with various concentrations of Olaparib as indicated in Fig. 2h for 6 days. Colonies were fixed with acetic acid/methanol and stained using 1% crystal violet in methanol, and hand counted. Experiments represent results from two independent experiments. For cell viability using the colorimetric MTS assay, cells ($1–2 \times 10^3$ cells) were seeded into 96-well plates for 24 h and treated with varying concentrations of cisplatin as indicated in Fig. 2i. After untreated control cells obtained ~80% confluence, the MTS assay was performed according to manufacturer's instructions (CellTiter 96 AQueous One Solution Cell Proliferation Assay, Promega). Experiments were performed in triplicates and repeated independently. Data was analyzed using Prism6 software and represents the mean +/− Standard error of the mean (SEM).

## Homology directed repair assay

Homology directed repair capacity was determined by transient integration of a reporter plasmid (gift from the Maria Jasin lab, Memorial Sloan Kettering) in HAP1 cells as previously described[61]. The cells were analyzed 72 h after transfection of DR-GFP and ISCE-I transfection using Lipofectamine 2000 (Invitrogen), by flow cytometry. See Supplementary Fig. 4b for gating strategy.

## Proximity ligation assay (PLA)

PLA experiments were performed according to the manufacturer's instructions. Briefly, cells were treated with 0.2 mM HU for 4 h where indicated and fixed with 2% PFA for 15 mins at RT. Subsequently, cells were permeabilized with PBS and 0.2% TritonX100 and blocked with PBS, 10% goat serum, and 0.1% TritonX100. Indicated primary antibody pairs were incubated overnight, followed by Duolink Sigma PLA reactions. Nuclear PLA signals were quantified for this analysis.

## In situ protein interactions with nascent DNA replication forks (SIRF)

SIRF assays were conducted as previously described[62]. Briefly, cells were pulse treated with EdU for 8 mins, followed by 0.2 mM HU for the indicated time. Cells were fixed, permeabilized and click-iT reaction was performed using biotin azide and AlexaFluor 488 azide according to manufacturers' instructions. Subsequently, PLA was performed using antibodies against the indicated RAD51C variant and biotinylated EdU. SIRF signals were quantified and normalized as previously described[62].

## Immunoblotting

Freshly harvested cell pellets were lysed in NETN buffer (50 mM Tris-HCl, pH 7.4, 1% NP-40, 200 mM NaCl, and 1 mM EDTA) on ice and underwent five freeze-thaw cycles. Lysates were centrifuged at $13,000 \times g$ for 15 mins and supernatant proteins were resolved by SDS-PAGE. Proteins were transferred to nitrocellulose membranes, blocked

with TBST and 5% milk and incubated with primary and corresponding secondary antibodies. Signals were detected using enhanced chemiluminescence.

## Immunoprecipitation

HAP1 cells were treated with 0.2 mM HU for 4 h at 37 C in 100 mm cell culture plates. Post treatment, ice cold RIPA buffer supplemented with protease inhibitors was added to cells (1 mL per plate) and cells were scraped off the plates. The cell suspension was rotated on orbital shaker for 15 min at 4 °C for complete lysis, and protein concentration was measured using the Bradford method. For the immunoprecipitation, cell lysates were incubated with indicated antibodies using protein A/G beads overnight. Post immunoprecipitation, the beads were washed with ice-cold PBS and proteins were eluted using BioRad Laemmli buffer. Proteins were resolved by SDS PAGE and immunoblotting was performed as described above.

## DNA fibers assays

DNA fiber experiments were performed as described previously[17]. Briefly, log-phase cells were pulse-labeled with 50 μM IdU and CldU with or without replication stress agents as indicated in figures. Cells were harvested, lysed, and spread to obtain single DNA molecules on microscope slides before standard immunofluorescence with antibodies against IdU and CldU (Novus Biologicals, BD Biosciences).

## Statistics and reproducibility

The statistical details of the experiments can be found in the figure legends in the article.

## Reporting summary

Further information on research design is available in the Nature Portfolio Reporting Summary linked to this article.

# Data availability

The Genbank accession codes for Alvinella pompejana RAD51C and XRCC3 are OQ586110 and OQ586109, respectively. The crystallographic models and data have been deposited in the protein data bank (PDB) with the following accession codes: 8GJ9 (apRAD51C N-terminal domain), 8GJ8 (apRAD51C C-terminal domain), and 8GJA (apRAD51C-XRCC3 core). The SAXS data and best-fit model has been deposited in the small-angle scattering biological databank (SASBDB) with the accession code SASDS36. Source data are provided with this paper.

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

## Acknowledgements

We thank Prof. Junjie Chen (MD Anderson) and Dr. Susan Lees-Miller (University of Calgary) for critical feedback to improve the manuscript. We thank Aditya Anand and Soumita SilDas (Lawrence Berkeley National Laboratory) for technical assistance on ApRAD51C experiments. This research was supported in part by the National Institute of Health (NIH) (1R01ES029680 to K.S., R35CA220430 to J.A.T., P01 CA092584 to G.J.W., D.S., and J.A.T.), Cancer Prevention and Research Institute of Texas (R1312, RP180463 to K.S.; RP180813 to K.S. and J.A.T.) and Robert A. Welch Chemistry Chair (J.A.T.), Canadian Institutes of Health Research PJT 159476 to G.J.W.). K.S. is a Rita Allen Foundation Fellow. K.S. and J.A.T. are CPRIT scholars in Cancer Biology. We acknowledge the Extreme Science and Engineering Discovery Environment (XSEDE, PSC allocations TG-BIO160040 and TGMCB170053), supported by NSF grant ACI-1548562, and the Texas Advanced Computing Center (TACC, http://www.tacc.utexas.edu) at The University of Texas at Austin for providing High Performance Computing (HPC) resources. Beamlines 8.3.1 and 12.3.1 at the Advanced Light Source are operated under support from NIH (P30 GM124169) and the Integrated Diffraction Analysis Technologies Program of the US Department of Energy Office of Biological and Environmental Research.

## Author contributions

K.S. and J.A.T. conceived the project. K.S. directed the study. K.S., J.A.T., and G.J.W contributed to design, interpretation, and supervision for experiments. M.A.L, G.J.W. did structural, biochemical, and in vitro binding experiments. A.S.A. contributed to X-ray diffraction data collection and model refinement. JTP contributed to SAXS data collection and analysis. A.B. did bioinformatic analyses. S.R. performed paralog interaction and SIRF studies, Y.C., S.K., R.A.B. performed DNA fiber experiments supported by K.S. for analysis, Y.C., C.K. performed cellular survival assay, K.H.T. performed xenograft and Dr-GFP experiments. D.S.

performed yeast-two-hybrid experiments, K.S., J.A.T. and G.J.W. wrote the manuscript.

## Competing interests

The authors declare no competing interests.
