## [Peer Review File · Nature Communications]

RAD51C-XRCC3 structure and cancer patient mutations define DNA replication rolesEditorial Note: This manuscript has been previously reviewed at another journal that is not operating a transparent peer review scheme. This document only contains reviewer comments and rebuttal letters for versions considered at *Nature Communications*. Mentions of prior referee reports have been redacted.

REVIEWERS' COMMENTS

Reviewer #1 (Remarks to the Author):

I am happy to support the revised manuscript. I thank the authors for their effort to address the comments. The paper will be a tremendous resource for the community, and it will guide future studies.

Minor point: something seems missing in this sentence from abstract: "Mapping of cancer patient mutations as a functional guide highlights unique CX3 interfaces plus ATP binding conserved with RAD51 recombinase. "

Reviewer #2 (Remarks to the Author):

RAD51C-XRCC3 structure and cancer mutations define DNA replication roles

The authors' rebuttal and alterations to the original manuscript largely address the majority of points raised by each of the reviewers. However, a few important major and minor points still remain to be addressed:

MAJOR:

Please consider altering the title, to remove "cancer mutations" as this is potentially misleading to the reader — 'recurrent patient mutations' may be a better descriptor.

Please provide a table listing each RAD51C mutation and its predicted / or confirmed effect on structure / function, whether it is known to be pathogenic or categorised as a VUS, plus appropriate citation of source.

Page 9. The additional text explaining the rationale of the mouse xenograft experiments is welcomed. However, it is still not clear as to why growth curves (Incucyte or similar) would not suffice to judge the effect of the introduced mutations on proliferation.

Page 9. "Taken together the data suggest that recurrent patient variants that cluster to two distinct CX3 interfaces in three dimensions show a functional defect despite HDR proficiency. Please add qualifying statement, along the lines of "... but this remains to be formally tested".

Page 18. Y2H data for A126T is required, and ideally supporting biochemical data showing that the A126T mutation affects the ability of the complex to bind ATP.

Please include XRCC3 in Figure 1a, clearly annotating the NTD- and CTD-interfaces (NTDI / CTDI).

Figures 3b/4d do not readily convey or communicate potential clashes between the mutated side chains and the rest of the protein.

MINOR

Page 5. Why is the polymerization motif (linker) of XRCC3 described as unique? Please expand.

Page 6. 'This data confirms that the apCX3 structure is conserved with the human complex in solution...' >> This data is consistent with apCX3 adopting the same structure as the human complex in solution.

Page 6. Subheading 'Cancer mutations at interaction regions' >> Recurrent patient mutations at interaction regions

Page 15. "... with RAD51C G264 exhibiting a much milder effect" >> "RAD51C G264S"

Reviewer #3 (Remarks to the Author):

Authors have made significant changes in the manuscript and have satisfactorily addressed most of the concerns. Addition of new data has strengthened their conclusions.

Review Response: "RAD51C-XRCC3 structure and cancer mutations define DNA replication roles"

For clarity *reviewer comments are in italic font*; responses are in blue standard font here and in the revised text.

We thank the editor and all three reviewers for their time in reviewing our manuscript and providing constructive feedback. By addressing the reviewers comments in our revised manuscript, we have improved clarity and incorporated points that increase and broaden the interest.

Referee #1:

I am happy to support the revised manuscript. I thank the authors for their effort to address the comments. The paper will be a tremendous resource for the community, and it will guide future studies.

Thank you! We also believe that our contributions in this paper will be a great resource for the community and our results and insights will guide future studies.

Minor point: something seems missing in this sentence from abstract: "Mapping of cancer patient mutations as a functional guide highlights unique CX3 interfaces plus ATP binding conserved with RAD51 recombinase."

We have modified this sentence in the abstract for clarity. The relevant sentence now reads "Mapping of cancer patient mutations as a functional guide confirms ATP-binding matching RAD51 recombinase, yet highlights distinct CX3 interfaces"

Referee #2:

The authors' rebuttal and alterations to the original manuscript largely address the majority of points raised by each of the reviewers. However, a few important major and minor points still remain to be addressed:

Thank you, we have addressed the remaining points to be addressed as detailed below:

MAJOR:

Please consider altering the title, to remove "cancer mutations" as this is potentially misleading to the reader — 'recurrent patient mutations' may be a better descriptor.

We have altered the manuscript title to "RAD51C-XRCC3 structure and cancer patient mutations define DNA replication roles". Recurrent cancer mutations is accurate, whereas using "patient" is not suitable because it is ambiguous in terms of what disease it is referring to.

Please provide a table listing each RAD51C mutation and its predicted / or confirmed effect on structure / function, whether it is known to be pathogenic or categorised as a VUS, plus appropriate citation of source.

We did not add the suggested table, as we respectfully suggest that it would derivative, speculative, and beyond the scope of this publication. Notably, here we report novel structures and combined functional analyses for Rad51C-Xrcc3 ATPase complex. The findings provide insights on exemplary mutations in cancer and Fanconi Anemia patients that combined with extensive functional experiments uncover distinct DNA replication fork protection, restart and reversal regions. However, the suggested table would largely duplicate cited published efforts on variant sites, and either be speculative (with unsupported functional predictions) or be a

major effort that goes far beyond the scope of one publication if we included the experimental tests we applied to the sites that we specifically studied.

Importantly, we have deposited the X-ray crystal structures and X-ray scattering structural data to make it a freely accessible resource worldwide for others to extend our studies on all other mutations (currently there are ~230 reported mutations). We specifically defined regions for distinct replication reactions, protein interactions and DNA binding. These delineated regions will now provide foundational information to guide appropriate functional tests for any given mutation within the identified regions. Regarding pathogenicity, the cited Prakash et al. provides a comprehensive list of many RAD51C mutations and their current predictions (<https://doi.org/10.1073/pnas.2202727119>, dataset S01). So adding a duplicate analogous table would needlessly undercut the published table, and as it would be derivative of what is already published, it would not be appropriate for *Nature Communications*.

Page 9. The additional text explaining the rationale of the mouse xenograft experiments is welcomed. However, it is still not clear as to why growth curves (Incucyte or similar) would not suffice to judge the effect of the introduced mutations on proliferation.

We have further clarified in the text that mouse xenograft experiments were used because they are more sensitive and biologically relevant assays than incucyte or similar.

Page 9. "Taken together the data suggest that recurrent patient variants that cluster to two distinct CX3 interfaces in three dimensions show a functional defect despite HDR proficiency. Please add qualifying statement, along the lines of "... but this remains to be formally tested".

We have modified this section of text for clarity such that it now reads "Taken together, these data suggest that recurrent patient variants that cluster to two distinct CX3 interfaces in three-dimensions show a functional defect despite HDR proficiency. These identified functional interfaces and their implied mutational defects now enable targeted testing of additional cancer patient variants for a comprehensive understanding."

Page 18. Y2H data for A126T is required, and ideally supporting biochemical data showing that the A126T mutation affects the ability of the complex to bind ATP.

We now specifically state in the manuscript that we do not have yeast two-hybrid data for A126T (although this is published in reference 3 of our manuscript) or G264S. Unfortunately, we have been unable to produce the CX3 complex containing the A126T mutation for biochemical assays despite extensive efforts, consistent with difficulties we and others have had in producing RAD51 paralog mutations for biochemistry. Notably we have worked many years to produce the data in this manuscript. The additional yeast two-hybrid data we added to Fig. 5 in response to reviewer 1 was collected by co-author David Schild in 2016, and he has since retired. We have therefore been unable to do additional yeast two-hybrid experiments to include added variants.

Please include XRCC3 in Figure 1a, clearly annotating the NTD- and CTD-interfaces(NTDI / CTDI).

Good point. We have added a schematic of XRCC3, and added labels for the NTD- and CTD-interfaces.

Figures 3b/4d do not readily convey or communicate potential clashes between the mutated side chains and the rest of the protein.

We have modified these figures to better highlight the major clash that the RAD51C G125V residue makes in Figure 3b. There are no major clashes between mutated residues in Fig 4d, although as stated in the text those mutations may lead to rearrangements at that interface.

MINOR

Page 5. Why is the polymerization motif (linker) of XRCC3 described as unique? Please expand.

We have added additional description about the XRCC3 polymerization motif and its comparison to both RAD51 and RecA to explain why it is unique. This includes an additional Supplementary figure 2e.

Page 6. 'This data confirms that the apCX3 structure is conserved with the human complex in solution... '>> This data is consistent with apCX3 adopting the same structure as the human complex in solution.

We have modified the relevant sentence to: "This data is consistent with the apCX3 core complex adopting the same structure as the human complex in solution with the RAD51C NTD flexibly attached, in concordance with our limited proteolysis results"

Page 6. Subheading 'Cancer mutations at interaction regions' >> Recurrent patient mutations at interaction regions

We have changed this subheading to "Cancer mutations at interaction regions".

Page 15. "... with RAD51C G264 exhibiting a much milder effect" >> "RAD51C G264S"

Thank you, we have fixed this typo.

Referee #3:

Authors have made significant changes in the manuscript and have satisfactorily addressed most of the concerns. Addition of new data has strengthened their conclusions.

Thank you!